# The S-component fold: a link between bacterial transporters and receptors
Michele Partipilo & Dirk Jan Slotboom ✉

The processes of nutrient uptake and signal sensing are crucial for microbial survival and adaptation. Membrane-embedded proteins involved in these functions (transporters and receptors) are commonly regarded as unrelated in terms of sequence, structure, mechanism of action and evolutionary history. Here, we analyze the protein structural universe using recently developed artificial intelligence-based structure prediction tools, and find an unexpected link between prominent groups of microbial transporters and receptors. The so-called S-components of Energy-Coupling Factor (ECF) transporters, and the membrane domains of sensor histidine kinases of the 5TMR cluster share a structural fold. The discovery of their relatedness manifests a widespread case of prokaryotic "transceptors" (related proteins with transport or receptor function), showcases how artificial intelligence-based structure predictions reveal unchartered evolutionary connections between proteins, and provides new avenues for engineering transport and signaling functions in bacteria.

Membrane-embedded sensory kinases transduce information across the lipid bilayer, while membrane transporters translocate molecules and ions. The two classes of proteins are generally considered unrelated, having evolved from different ancestral proteins[1,2]. In bacteria, signal sensing is often mediated by the two-component systems consisting of a multi-domain sensor histidine kinase (SHK) that binds the signaling molecule and initiates a phosphorylation cascade, and a response regulator that orchestrates gene regulation[3]. In contrast, transported nutrients are recognized by unrelated proteins that undergo a series of conformational changes leading to molecular translocation across the membrane[4]. Although operated by distinct entities, transport and signal sensing can function synergistically, as demonstrated by the complex formed by the C4-dicarboxylate transporter DctA and the fumarate sensor DcuS in *Escherichia coli*[5], or as shown with the structural characterization of the BceAB-S module from *Bacillus subtilis*[6], where an ABC transporter (BceAB) interfaces an histidine kinase (BceS) to respond to antimicrobial peptides.

Over the past decades, rare cases of membrane proteins named 'transceptors'[7–9] have been reported indicating that this distinction between receptors and transporters may not be absolute. We here define transceptors strictly as membrane folds, thus excluding scenarios in which a dedicated soluble domain acts as a bridge between transport proteins and signal transduction systems (e.g., STAC domain alone or incorporated into CbrA from *Pseudomonas putida*)[10,11]. The identified membrane transceptors structurally resemble transporters, yet function as receptors, suggesting that

the two processes might be evolutionary linked. However, based on the extremely low number of reported cases, and their exclusive presence in eukaryotic organisms (detailed list in Supplementary Table 1)[12–19], transceptors appear to be oddities. The apparent rareness is further underscored because each known transceptor case constitutes only a single protein in a large transporter family that is likely to have evolved from a transporter to a receptor function.

Sequence comparisons have not been able to detect a more widespread occurrence of transceptors across the domains of life, but it is well known that protein structures are better conserved than amino acid sequences, and therefore better suited to find distant evolutionary relations[20]. In a recent study, a new transceptor was found when the structure of a KDEL receptor, involved in intracellular trafficking of eukaryotic proteins, was solved. The protein turned out to be a structural homolog of the sugar transporters of the SWEET family, despite the lack of appreciable sequence identity between these membrane proteins[21]. While structural similarity searches may be used to find more transceptor cases, the limited availability of experimentally determined structures often makes it impossible to uncover such relations. Computational structure prediction could potentially fill this gap, yet until recently its lack of accuracy has hindered successful use in protein structure determination. The emergence of AlphaFold[22,23], and the resulting millions of accurately predicted protein structures, including those of integral membrane proteins[24], now allow to search for proteins structurally related to transporters and receptors that were previously out of the reach of conventional structural biology.

Department of Biochemistry, Groningen Institute of Biomolecular Sciences & Biotechnology, University of Groningen, Nijenborgh 4, 9747 AG Groningen, The Netherlands. ✉e-mail: d.j.slotboom@rug.nl

Here, we used AlphaFold structures to search for transceptors in prokaryotes. We focused on bacterial ATP-binding cassette (ABC) transporters of type III, the so-called Energy-Coupling Factor (ECF) transporters[25,26]. ECF transporters consist of a three-subunit motor module (termed ECF module) that binds and hydrolyzes ATP, and an integral membrane subunit (S-component or EcfS) that binds and translocates the cognate substrate across the membrane upon association with the motor module (Fig. 1)[27,28]. While the ECF module is well conserved with multiple typifying motifs (i.e., Walker A and B motifs, signature motifs of the ATPase subunits, and Ala-Arg-Gly motifs of the scaffold protein EcfT)[26,29], S-components with specificity for different transported substrates display highly variable sequences, which has complicated the identification of the encoding genes. Nonetheless, all structurally characterized S-components have a conserved characteristic fold with six membrane embedded helices. We used Alphafold structure predictions to identify previously undetected S-components. Surprisingly, we found that S-components are structurally similar to the membrane domains common to a large number of sensor histidine kinases (the so-called 5TMR-SHKs) and chemotactic receptors, suggesting an evolutionary link between these groups of proteins. Thus, we hypothesize that SHKs might have evolved from ancestral transporters that gained the new function of redirecting the metabolic response without translocating their ligand or vice versa. We show that the S-component fold represents the most widespread manifestation of transceptors to date, and the first case found in prokaryotes. We also conclude that the S-component fold is much more abundant and versatile than previously thought, and can specialize in multiple functions, including molecular transport and signal transduction.

## Results
### The fold of S-components is conserved in proteins with different function from micronutrient import
S-components from ECF transporters, specific for different substrates, all share a characteristic fold consisting of six membrane-spanning α-helices, but they do not exhibit significant sequence identity (~15%)[26], a feature that makes the identification of related proteins based on sequences non-trivial[30]. We used the Foldseek webserver[31] to search for proteins structurally related to S-components in the databases of protein structures predicted by the AlphaFold neural network. As input for the search, we took experimentally determined structures of four S-components: RibU[32] from *Staphylococcus aureus* TCH60, FolT[33] from *Lactobacillus delbrueckii* ATC 11842, ThiT[34] and BioY[35] from *Lactococcus lactis* MG1363 – that mediate transport of riboflavin, folate, thiamine and biotin, respectively. In parallel, we also used

the AlphaFold predicted structures of the same four proteins as input in our searches. The search in the AlphaFold model databases AFDB proteome and AFDB-SwissProt retrieved proteins with the same fold as the queries (Fig. 2a). Unsurprisingly, a substantial portion of the annotated hits are other S-components (PFAM entry: PF12822), regardless of the predictive or experimental nature of the S-component structure used as entry. In all cases, the best-scoring matches were homologous S-components with identical predicted substrate specificity to the protein query. Unexpectedly and consistently, we also found proteins functionally unrelated to S-components. The most abundant ones are annotated as sensor histidine kinases (SHKs, PFAM entry: PF07694), specifically the pyruvate sensors YpdA and BtsS, and the sporulation kinase B KinB. In the past, before accurate structure predictions were possible, these histidine kinases had been named 5TMR-LYT (5 TransMembrane Receptor of the LytS-YhcK type)[36], a name that is now known to be incorrect, because the proteins contain more than five transmembrane domains (discussed in detail below)[37]. In addition, we found rod-shape determining proteins of the MreD family (PFAM entry: PF04093), which had already been suggested to have the S-component fold[38]. SHKs constituted a larger fraction of the hits (up to 25% of hits in the AFDB-proteome database, depending on the query protein) than MreD proteins, most likely because the sole function of MreD is tightly coupled to determining cell architecture in rod-shaped bacteria[39–42], while signal sensing receptors have diversified to respond to different stimuli[43]. Finally, entries not functionally annotated represented 20 and 60% of the hits, depending on the specific search, and in most cases their AI-predicted structure suggests the function of S-components (although not annotated as such in protein databases). Both for the SHK and MreD hits, the structural predictions were of high confidence (Supplementary Table 2). To validate AlphaFold-predicted structural similarity between S-components and 5TMR-SHKs, we used well-established approaches of protein fold recognition[44–46] using the sequence of the membrane domain of 5TMR-SHKs (Supplementary Tables 3 and 4): crystal structures of S-components were always scoring among the high-confidence hits for structural relatedness with SHKs, consistently with the similarity revealed from the AI-based tools.

The core of the shared fold, which we refer to as "S-component fold", is composed of six alpha-helices (H1–H6) connected by 5 loops protruding alternately towards the extracellular side and the cytoplasm (Fig. 2b–d)[47,48]. The 5TMR domain of SHKs has an additional helix at the N-terminus (H0) linked to the six-helical core by a cytosolic loop, resulting in a total of 7 transmembrane helices, as recently demonstrated biochemically for the high-affinity pyruvate receptor BtsS[37]. H0 has its N-terminus on the

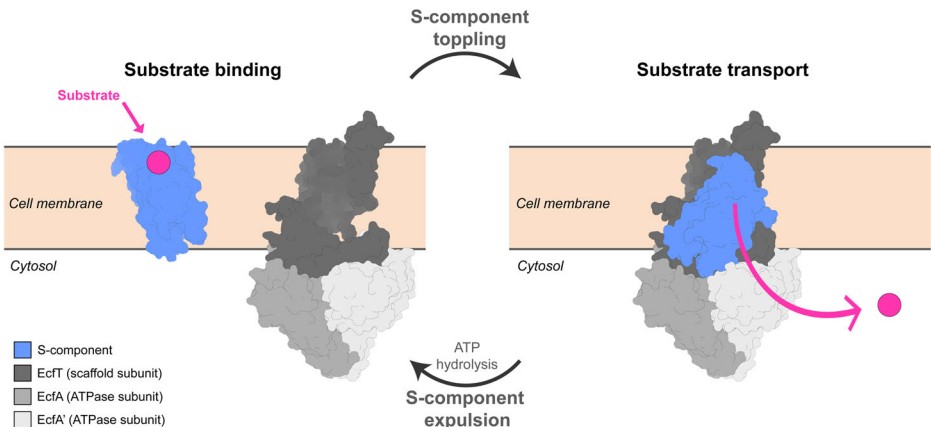

**Fig. 1 | General scheme of the transport cycle of Energy-Coupling-Factor (ECF) transporters.** An S-component (in light blue) that bound its specific substrate on the extracellular side, interacts via a toppling mechanism with a tripartite motor module, composed of two soluble ATP-hydrolyzing domains (EcfA and EcfA' in light shades of gray) and a transmembrane scaffold subunit (EcfT, in dark gray). The association in a full complex leads to substrate translocation into the cytosol. The

binding and subsequent hydrolysis of ATP by the ATPase dimer initiate a sequence of conformational changes culminating in the expulsion of the S-component from the ECF module. This resets the module, making it available once again to associate with a new substrate-bound S-component for another transport cycle. Deformations of the lipid bilayer mediating the toppling mechanism[68] are here omitted.

**Fig. 2 | The S-component fold in membrane proteins unrelated to molecular transport. a** Top hits from the structural similarity search via FoldSeek from S-components with different substrate specificity. The AFDB-proteome database and the AFDB-SwissProt database were searched using both experimentally determined (PDB entries: 3P5N, 5D0Y, 4POP, 4DVE) and AI-predicted structures (PDB entries: AF-E5QVT2-F1, AF-Q1G930-F1, AF-A2RI47-F1, AF-A2RMJ9-F1) from the Alpha-Fold tool (v2.3.2). The data represent the hits with a probability equal or higher than 50% of sharing a protein fold identical to either RibU, FolT, ThiT or BioY. **b–d** AlphaFold structures of RibU **b**, YpdA **c** and MreD **d** sharing the same fold of six membrane-spanning α-helices (H1-H6, in rainbow). The structure identifiers are AF-Q8Y5W0-F1, AF-P0AA93-F1 and AF-Q2FXS7-F1, respectively. The membrane domain of 5TMR-sensor histidine kinases exhibits an additional helix H0 at the N-terminus (in magenta), while the C-terminus expands intracellularly into soluble domains (in grey) for signal transduction, here shown for the specific case of YpdA.

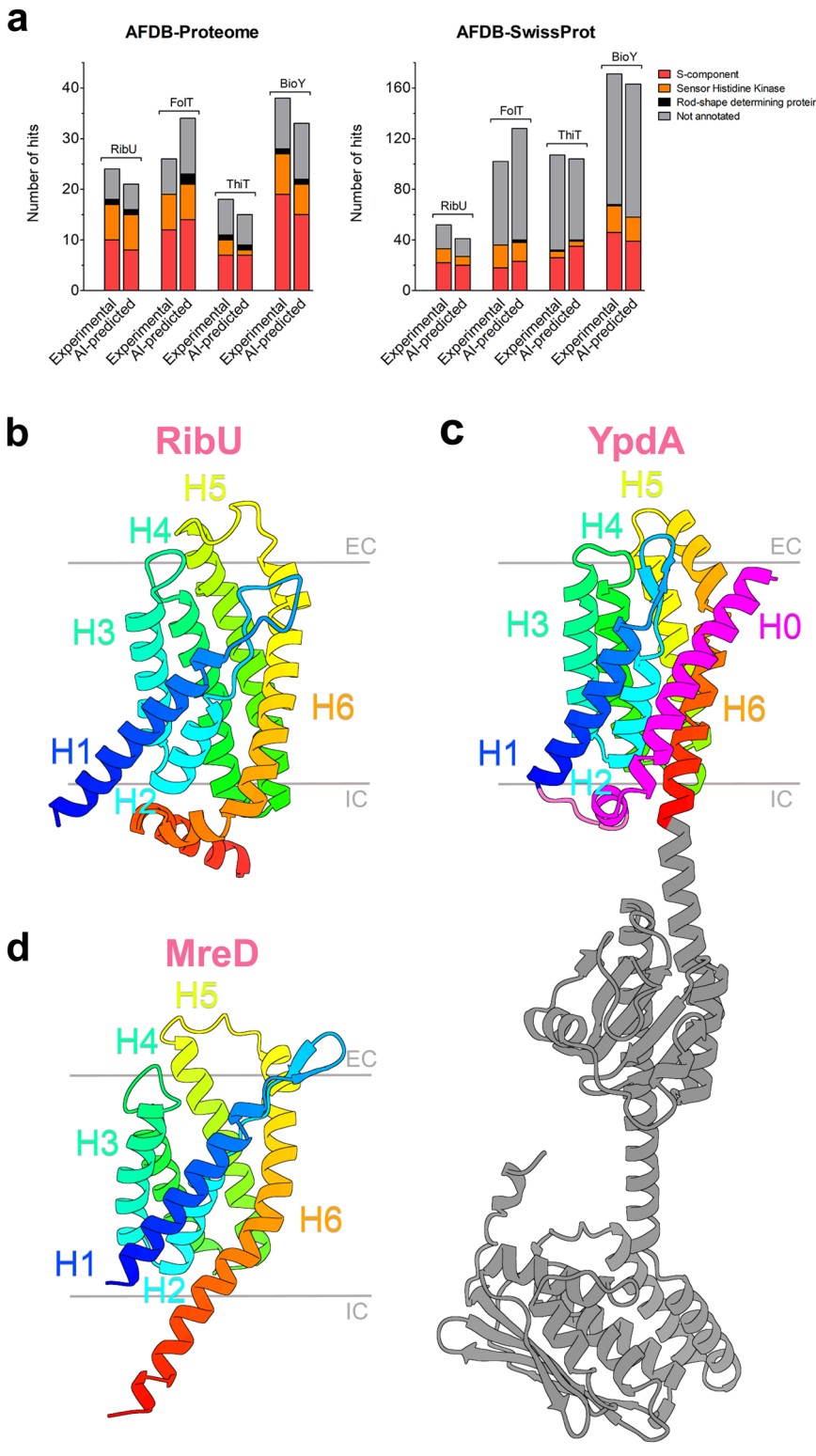

extracellular side, and the C-terminus at the cytoplasmic side of the membrane. H6 delimits the C-terminal end of the protein in the case of S-components and MreD, but not of the SHKs, which instead extend from the membrane part towards the cytoplasm into soluble domains required for signal transduction[49]. Because the exact role of MreDs is unclear, and mechanistic insight in rod-shape determination is lacking, we focus primarily on the relation between S-components and SHKs, and discuss MreDs only in the context of phylogenetic distribution.

## The shared fold is widespread across bacterial phyla

S-components of ECF transporters are found in some archaea, but are much more prevalent across bacteria[50] (Fig. 3a). A similar distribution was observed for the SHKs with shared S-component fold, while MreDs are entirely absent in Archaea. This aligns well with a crenactin-based cytoskeleton (rather than employing MreB, MreC, and MreD proteins)[40] responsible for shape determination in most rod-shaped archaea[51].

**Fig. 3 | Taxonomic distribution of the shared six-helical fold. a** The occurrence among different species was obtained from the respective PFAM entries available in the 'Taxonomy' section of the InterPro database https://www.ebi.ac.uk/interpro/entry/pfam/ (PF12822 for S-components, PF04093 for MreD proteins and PF07694 for 5TMR-containing SHKs). **b** The distribution of S-components, SHKs and MreDs among bacterial phyla. The analysis on the number of species refers to the release InterPro 96.0.

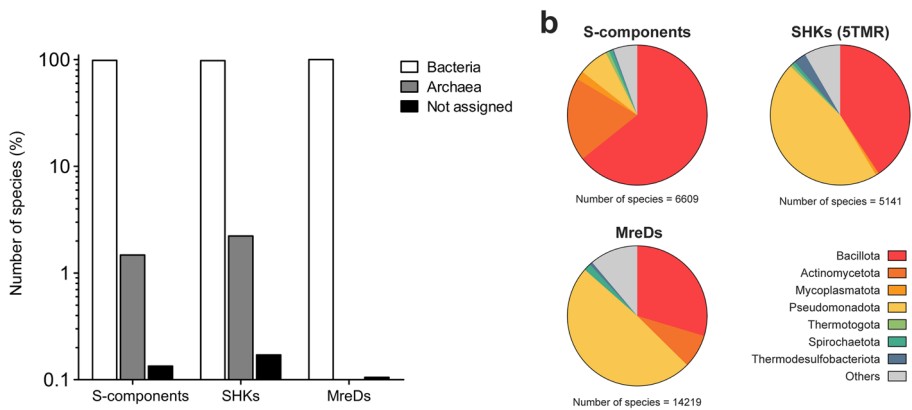

Among bacteria, S-components of ECF transporters have predominant presence in Gram-positive microorganisms: more than 75% of known S-components were found in Bacillota and Actinomycetota[25,26] (Fig. 3b). A very different picture emerges for the SHKs and MreDs, as they both prevail in the Pseudomonadota phylum, constituting 46% and 49% of the respective total distributions, but are also well represented on Gram-positive bacteria (41% and 37% of the total). Among Gram-positive microbes, SHKs with the S-component fold are found almost exclusively in Bacillota, while MreDs are also present in Actinomycetota. Collectively, we can conclude that the shared S-component fold is diverse not only in regard of the functional specialization but also for widespread occurrence across microorganisms with different morphologies and metabolisms.

In the absence of sequence similarity, common function or comparable taxonomic distributions, we tried to unravel the relationship between the protein groups sharing the S-component fold by taking advantage of their AI-predicted structures. The AlphaFold structures of representative proteins with the S-component fold from ECF transporters, SHKs and MreDs were used for structural multi-alignments and to build a phylogenetic tree (Supplementary Fig. 1 and Supplementary Table 5). However, the resolution of the tree based on the structural alignment was not high enough for definitive conclusions about the evolutionary origin of the fold. In future, experimental structures, improved structural predictions and AI-based structural phylogenetics may overcome this limitation[52,53].

### Fold specialization of the 5TMR-containing sensor histidine kinases

We reasoned that the previously proposed role of the N-terminal helix H0 of SHKs as signal peptide was potentially incorrect since its membrane topology (with the N- and C-termini on the extra- and intracellular sides, respectively) is opposite to the canonical configuration in which the cleavage site downstream of the helix is located extracellularly (Fig. 2c)[54]. Consistently, the predicted probability of a signal peptide sequence in this helix by the algorithm SignalP 5.0 was virtually zero (Supplementary Fig. 2). Instead, the H0 domains of SHKs contain well conserved regularly spaced leucine residues (Fig. 4a), reminiscent of leucine zipper-like motifs, a feature that in the case of sensor histidine kinases might facilitate dimerization within the bilayer, which is required for the receptor functionality[55]. In addition, the conservation of two consecutive polar amino acids located towards the extracellular side of the alpha helix (shaded in orange in the Fig. 4a) further supports the potential function of helix H0 in promoting dimerization[56,57]. AlphaFold predictions for the dimeric conformation of 5TMR-SHKs highlighted the close proximity – and probable interaction - of the conserved polar side chains (around 4 Å), corroborating the hypothesis of the formation of an intramembranous salt bridge between two protomers.

A further conserved sequence characteristic of the 5TMR domain of SHKs is a NXR motif located between helix 1 and helix 2, towards the extracellular side of the membrane[36]. The side chains of the asparagine and arginine residues point toward the central cavity delineated by the

transmembrane helices, where substrate binding takes place in the S-components of ECF transporters[32,58]. Therefore, the two residues in the 5TMR domain may play a role for the binding of the external signaling molecule. Four additional conserved motifs face the same cavity, all of which located at positions that in S-components are involved in substrate binding (Fig. 4b, and Supplementary Figs. 3 and 4). The conserved residues are distributed over multiple helices of the core fold, suggesting that substrate binding leads to intramolecular conformational changes, similar to what has been shown for S-components of ECF transporters. While in ECF transporters, the binding of substrate leads to association with the ECF module, we speculate that in SHKs, conformational changes upon substrate binding may lead to a specific reorganization of the pre-existing interactions between the two receptor protomers, and the transmission of the signal from the membrane to the soluble domains.

### Intracellular domains of 5TMR-SHK receptors

A limited number of combinations of intracellular domains has been reported to be linked to the 5TMR (S-component-like) membrane domain of SHKs[36], including GGDEF and EAL domains (producing and degrading cyclic-di-GMP)[59], PAS and GAF domains (involved in receptor dimerization and binding of cyclic nucleotides)[60,61], and HK and HK-ATPase domains (synergically responsible for signal transduction in numerous histidine kinases)[62]. Analysis of the much larger group of 5TMR-SHKs (PFAM entry: PF07694) found by the structure prediction revealed that the soluble domains are much more diverse (Fig. 4c, and Supplementary Table 6), reaching almost 500 different functionally annotated combinations. Although the soluble PAS, GGDEF and EAL domains are often fused with the 5TMR domain, they can be combined in a modular manner. While YpdA falls into the category of 5TMR-SHKs with a GAF domain separating the membrane fold from the soluble HK and HK-ATPase domains, variations of the classic configuration with only HK and HK-ATPase also are present. They have one or more response regulator receiver (REC) domains fused to the C-terminus of the HK-ATPase of the receptor, thus forming the so-called hybrid histidine kinases[63,64], which are specialized in the multistep phosphorelay within the receptor or towards other response regulator proteins.

Surprisingly, we also discovered three unprecedented domains fused to the 5TMR fold namely methyl-accepting chemotaxis protein (MCP) signaling domain (PFAM entry: PF00015), PPM-type phosphatase-like (annotated as SpoIIE in the PFAM entry: PF07228), and natural resistance-associated macrophage protein (NRAMP)-like (PFAM entry: PF01566). The association with a MCP domain strongly suggests that the substrate recognized by 5TMR-receptors can directly trigger a chemotactic response in the microorganisms equipped with such a domain architecture (Supplementary Table 7). In some cases, a PAS domain separates the 5TMR fold from the MCP domain, mostly in bacteria belonging to the order *Hyphomicrobiales*, but there are different cases where the membrane and the chemotactic domains are directly attached by a short coil (e.g., in certain

**Fig. 4 | Structural insight into the sensor histidine kinases containing the 5TMR domain. a** The function of helix 0 (H0) in the membrane domain. The sequence logo of the N-terminal transmembrane helix H0, was obtained from the multiple alignment of sequences reported in the PFAM entry 5TMR-LYT (PF07694), and generated with WebLogo[87]. Amino acids in the logo sequence are colored according to their hydrophobicity (hydrophobic in black, hydrophilic in blue and neutral in green). The polar amino acids proposed for a potential transmembrane salt-bridge in the dimeric conformation of the receptor are shaded in orange. On the right, the AlphaFold predicted dimeric conformation of 5TMR-SHKs generated with ColabFold[88] is shown using YpdA from *Escherichia coli* K-12 as a template. H0 is colored in magenta, the cytosolic loop linking H0 to the six-helical membrane fold is in white, the 5TMR domain is highlighted in blue. **b** Conserved amino acids potentially involved in substrate recognition. The color scheme of amino acids is the same as described in panel a, the numeration refers to the sequence of YpdA from *E. coli*. Transmembrane helices are numbered H1-H6, while H0 is highlighted in magenta. **c** Diversity of the cytosolic domains attached to the 5TMR domain. Some examples of this modular assembly are represented from membrane receptors present in both bacterial and archaeal organisms, with the respective UniProt codes indicated above the corresponding AlphaFold structure. The legend identifying the cytosolic protein domains is reported at the bottom of the panel.

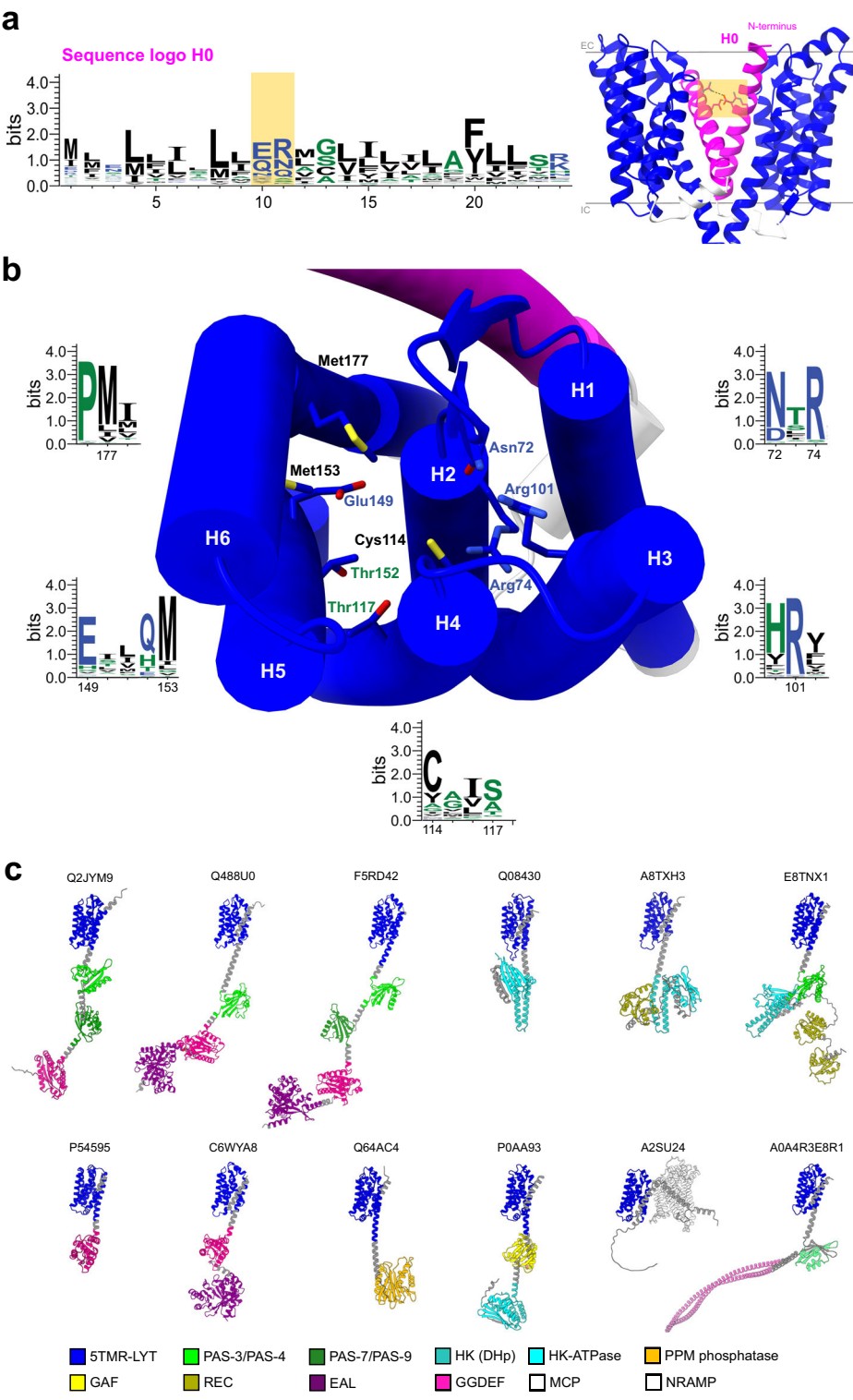

species of *Clostridium* and *Treponema*). The PPM-type phosphatase and the NRAMP domains are prevalent in archaea rather than in bacteria. The PPM domain was already known in the context of archaeal signal transduction[65,66]. The fusion of the sensorial 5TMR domain with NRAMP-like proteins constitutes instead a more exceptional scenario: not only has this architecture the 5TMR fold at the C-terminus of the protein sequence, but it also represents the only protein assembly in which an integral membrane transporter is used as a fused domain. NRAMP proteins are membrane transporters for divalent metals[67], potentially indicating a unique

metabolic specialization bridging ion transport and signal sensing to ensure intracellular homeostasis.

**The assembly with an ECF module for transport is associated with the conservation of the AXXXA domain in helix 1**

S-components from ECF transporters are specialized in binding their substrate with high affinity, and enable the translocation of substrate upon the association with a tripartite ATPase ECF module[27,28]. The association leads to a characteristic toppling motion in which the S-component rotates

**Fig. 5 | The divergence from the transport function of the fold of S-components is associated with the inability to form a complex with the ATPase energy-coupling module. a** The organization of Energy-Coupling-Factor (ECF) transporters. The pink shaded area highlights the multiple contact points between S-component and the subunits of the ECF tripartite module. **b** The conservation of the AXXXA motif in helix 1, involved in the transport function of the shared membrane fold. The sequence logos illustrate the amino acid frequency corresponding to the highlighted sequence of helix 1 (red rectangle) reported for S-components, 5TMR-containing SHKs and MreD proteins on https://www.ebi.ac.uk/interpro/entry/pfam/.

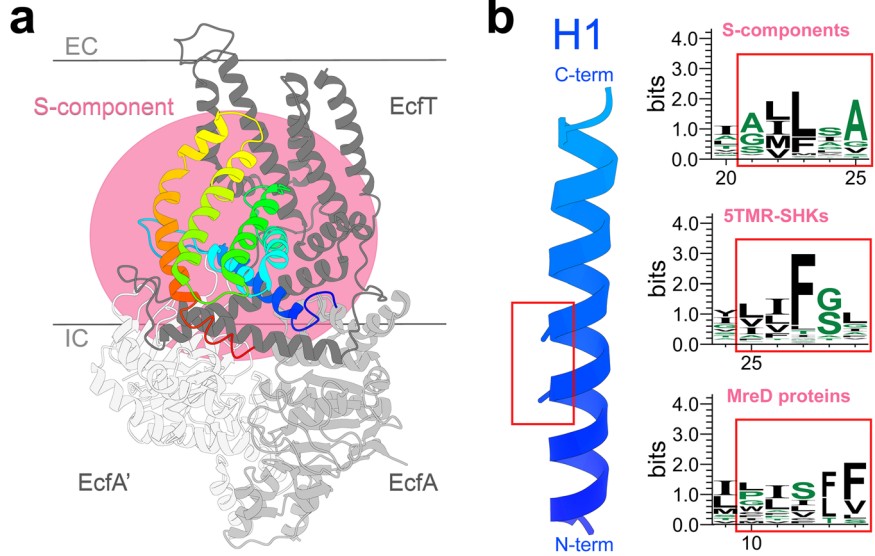

relatively to the membrane plane, which brings the substrate binding site from an outward- to an inward-facing orientation, allowing substrate release into the cytoplasm. The assembly of the S-component and the ECF module into a full ECF transporter (Fig. 5a) involves an AXXXA motif in helix 1, which is relatively conserved among S-components, as evidenced by the structures of the full complexes determined to date[32,68,69]. This motif is responsible for the tight association of the S-component with the scaffold protein of the ECF module (EcfT). Mutation of either alanine prevented the S-component ThiT to interact with the ECF module, and cause a complete loss of transport ability[70]. This typifying motif in helix 1 is absent from MreDs and SHKs (Fig. 5b). Instead, amino acids with large hydrophobic (leucine, isoleucine, valine) or aromatic (phenylalanine, tryptophan) side chains mostly occur at the positions of the two alanines, suggesting their inability to interact with the motor module of ECF transporters, and consistent with the different roles of MreDs (scaffold needed for cell shape determination)[42] and 5TMR-containing SHKs (signal sensing)[71].

## Discussion

The advancement of structure prediction and analysis tools based on artificial intelligence training (i.e. AlphaFold and Foldseek) is transforming life sciences subfields[72,73]. Combining these AI-based tools with bioinformatic approaches, we found that two major groups of bacterial membrane proteins share a core fold: S-components of ECF-type ABC transporters for micronutrient uptake and 5TMR-SHK receptors of two-component systems involved in signal sensing and chemotaxis. This structural relatedness between transporters and receptors manifests the first transceptor case identified in prokaryotes (receptors structurally related to transporters but not functionally), and the most abundant one to date. The evolutionary origin of the shared fold described in this work remains elusive, as we could not build conclusive phylogenetic trees from the AI-based structures currently available. The improvement of the computational methods, corroborated by experimentally determined structures and biochemical studies in vivo and in vitro[52], will allow to shed further light on the evolutionary origin of the bacterial transceptor here reported.

In the absence of experimentally determined structures for 5TMR-SHKs, the structural relatedness between S-components and these receptors must await experimental confirmation. In fact, the structures predicted by AlphaFold do not always correspond with those determined by conventional techniques for structure determination[52]. However, the confidence of our approach is high for multiple reasons: (i) we used both AI-predicted and experimentally determined structures of S-components to search for structural homologs (Fig. 2a), (ii) the TM-scores obtained for these protein folds of approximately 200 amino acids were in most cases above 0.5 (Supplementary Table 2), (iii) the corresponding RMSD values were mostly comprised between 4 and 7 Å (Supplementary Table 2), (iv) fold recognition from the primary sequence of the membrane domain of the 5TMR-SHKs always gave crystallographic structures of S-components (Supplementary Tables 3-4), (v) certain amino acids well conserved in the pocket for substrate recognition in 5TMR-SHKs (Fig. 4b) correspond to amino acids involved in substrate binding in S-components with different specificities (Fig. S4).

The S-component fold may be suitable for both transport (in the context of the ECF module) and signal transduction function for two reasons. First, when S-components are not associated with the ECF module, the substrate binding sites are located close to the extracellular side of the membrane. This location is also suitable for receptors, as for instance highlighted by G-protein-coupled receptors in eukaryotes[74]. Comparison of the AlphaFold structures of 5TMR-SHKs with the crystal structures of S-components in complex with their substrates, supports a similar position of the substrate binding sites (Supplementary Fig. 4): the NXR motif, Cys114 in helix 4 and Met177 in helix 6 of SHKs are equivalent to amino acids interacting with the cognate ligand in S-components (Fig. 4b). This proposed location of the ligand binding site is consistent with mutagenesis by Qiu and coworkers[37] of the pyruvate sensor BtsS.

Second, we speculate that analogously to S-components of ECF transporters (Fig. 6), substrate binding to 5TMR-SHKs may also initiate a toppling-like movement of the membrane fold. However, in 5TMR-SHKs the motion cannot be completed because the proteins do not interact with an ECF module, and it is counteracted by H0 and the cytoplasmic extensions of H6. Rather than toppling and associating with the ECF module, the conformational changes induced by the substrate recognition at the membrane domain interface may culminate into an orchestrated series of intramolecular rearrangements of the cytosolic domains, and eventually enable the downstream phosphorelay cascade. In the proposed step of ancestral fold specialization, the particular adaptation of the regions involved in the conformational changes towards the intracellular portion might have determined the specialization into either substrate translocators or signal sensing proteins. It is noteworthy that a few S-components have been found to be able of transporting a substrate across the membrane without the need for an ECF module (biotin by some BioY proteins and vitamin B12 by BtuM)[30,75]. These proteins might be considered candidate intermediates between ECF transporters and SHKs, although the mechanism that these solitary S-components use for transport is still unclear.

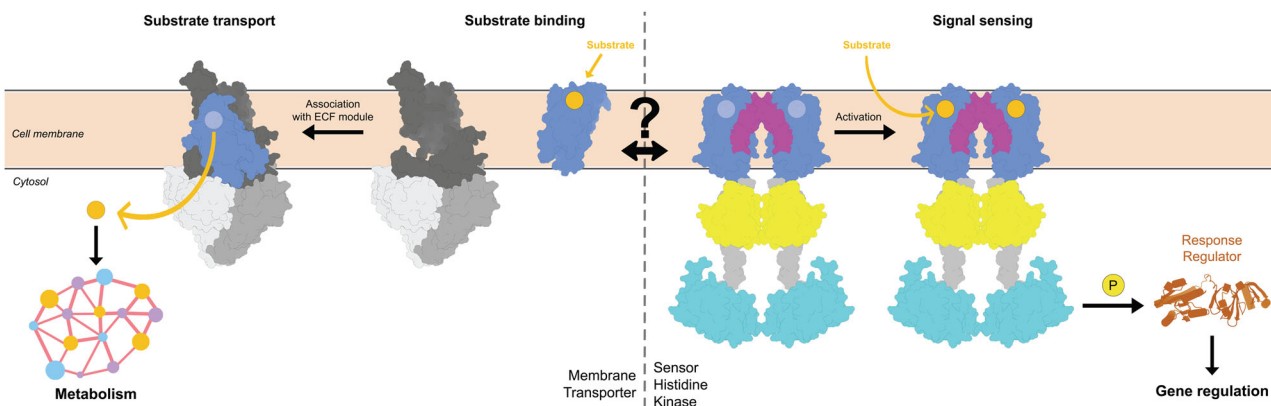

**Fig. 6 | Proposed divergent specialization of the shared fold into the transport and receptor functions.** In S-components (on the left), substrate binding initiates a sequence of intramolecular changes that prompt the protein to undergo an initial rotation within the membrane. The toppling movement terminates upon the association with the ECF motor module, which enables substrate delivery to the cellular metabolic machinery. Subsequent binding and hydrolysis of ATP resets the system. In 5TMR-sensor histidine kinases (on the right), the binding of the substrate initiates similar intramolecular conformational changes, but toppling is limited by the extension at the C-terminus into soluble domains. Following conformational changes involving both the membrane interface (helix 0 at the N-terminus in magenta, 5TMR-fold in blue) and the soluble domains, the receptor can transduce the signal via phosphorylation of the response regulator protein, which can eventually affect the gene response.

Our work may inspire engineering efforts to transform receptors in transporters and vice versa, with microbial manufacturing as main application[76]. Alternatively, the exclusive occurrence of S-components and sensor histidine kinases in bacteria can be exploited as common targets to design novel antimicrobial compounds inhibiting either micronutrient uptake or signal sensing and transduction in pathogenic strains[77,78], or both.

## Methods
### Structural similarity search
The entries of RibU (UniProt accession number E5QVT2), FolT (UniProt accession number Q1G930), ThiT (UniProt accession number A2RI47) and BioY (UniProt accession number A2RMJ9) from the protein database UniProt[79] (UniProt release 2023_04) were employed to obtain both experimentally determined (PDB entries: 3P5N, 5D0Y, 4POP, 4DVE) and AlphaFold-predicted structures (PDB entries: AF-E5QVT2-F1, AF-Q1G930-F1, AF-A2RI47-F1, AF-A2RMJ9-F1). The 3D maps were used as input entries for the structural similarity search on the Foldseek webserver[31] (release 8-ef4e960), and the resulting top hits were extracted from the AlphaFold model databases AFDB proteome and AFDB-SwissProt (Fig. 2a). The probability cut-off value for including an entry among the top hits with identical protein fold was set to 50%. In the case of experimentally determined structures, only the hits from chain A were considered for further analyses. Details on the structural similarity results obtained with YpdA. BtsS and KinB (e.g., template modeling score and root mean square deviation value) are available on Supplementary Table 2. The AlphaFold structures showing the shared fold (Fig. 2b–d) were visualized using UCSF CHIMERAX software (version 1.6.1)[80].

The amino acid sequences corresponding to the membrane domain (first 200 amino acids) of well-known 5TMR-SHKs[36,37,81,82] (UniProt accession numbers: P94513, P54595, P0AA93, P0AD14, Q08430, Q2G061) were used as entries to generate sequence–structure alignments via the PSIPRED workbench[46], and find crystallized structures with the same protein fold. The output of the fold prediction and protein family classification searches from the pGenTHREADER and pDomTHREADER methods[44] are reported on Supplementary Tables 3 and 4.

### Protein fold taxonomic distribution analysis
S-components and related protein hits from the structural similarity search were searched on the UniProt resource under the entry 'Family and domain databases'. The PFAM database, now integrated in the InterPro webserver (release InterPro 96.0)[83], returned as results the entries PF12822 (ECF transporter, substrate-specific component), PF07694 (5TMR of 5TMR-LYT) and PF04093 (rod shape-determining protein MreD), which were then consulted on the corresponding 'Taxonomy' section. For each entry, protein occurrence across domains of life (Fig. 3a) and bacterial phyla (Fig. 3b) were displayed in terms of number of species in percentage or absolute numbers, respectively. The distributions were represented using GraphPad Prism 9.0.0.

### Bioinformatic analyses on the shared fold
The primary structures of the abovementioned well-known 5TMR-SHKs were used to query the SignalP 5.0 webserver[84] with the aim to predict any signal peptide sequence at the N-terminus of the proteins (Supplementary Fig. 2). The sequence of RibU was used as a negative control, while that of GlnP[85] (UniProt accession number: Q9CES5) was used as a positive control. The inquiry on SignalP 5.0 accounted for the Gram-positive/negative origin of the organism harboring the receptors.

The sequence logo of helix 0 (Fig. 4a, on the left) was generated from the multiple sequence alignment of 5TMR-SHKs, performed on ClustalOmega[86], and consequently visualized via WebLogo[87]. The sequences used for the multialignment were collected from the InterPro resource, more precisely from the 'Alignment' section (https://www.ebi.ac.uk/interpro/entry/pfam/PF07694/entry_alignments/?type=seed), and were implemented with the sequences of the 5TMR-containing SHKs previously employed for the signal peptide prediction. To ensure that the alignment involved only the 5TMR domain, we excluded sequences in which the 5TMR domain is at the C-terminus (proteins with the NRAMP-1 in their domain architecture), and cut the sequences to just the first 180 amino acids.

The conserved regions of the 5TMR-containing SHKs (Fig. 4b) were highlighted from the multialignment of the same sequences mentioned above in the alignment, but involving exclusively the 5TMR domain (the amino acids corresponding to helix 0, cytosolic loop linking H0 to the shared fold, and intracellular domains were excluded). Also in this case, the visualization of the sequence logo was carried out via WebLogo. The top view of the six-helical fold localizing the conserved motifs of the localization of the conserved regions was generated by UCSF CHIMERAX software.

The dimeric conformation of 5TMR-receptors (Fig. 4a, on the right) was obtained using the ColabFold suite (version 1.5.2)[88] available on the web-based gateway COSMIC2[89]. Specifically, the primary sequence of YpdA from *E. coli* was employed as a template for the prediction of the homodimeric state selecting standard parameters (5 models, 3 recycles and stop at score of 80). Upon verification of the models, the predicted multimeric state was visualized via CHIMERAX.

The multiple combinations of sensor histidine kinases containing the 5TMR-domain (Fig. 4c) were retrieved from the 'Domain architectures' entry available on InterPro (https://www.ebi.ac.uk/interpro/entry/pfam/PF07694/domain_architecture/). Representative AlphaFold structures of SHKs showcasing the variegated intracellular assembly of the receptors were downloaded and visualized with CHIMERAX.

The ECF transporter shown in Fig. 5a corresponds to the folate ECF-transporter (PDB entry: 5D3M)[33] and its cartoon was generated using CHIMERAX, as well as the helix 1 of the respective S-component represented in Fig. 5b. The sequence logo of helix 1 (Fig. 5b) was obtained for each protein family (entries PF12822, PF07694, PF04093) from the multiple sequence alignment available on the InterPro resource at the correspondent 'Alignment' entry, selecting the sequences provided by the SEED database[90].

## Reporting summary
Further information on research design is available in the Nature Portfolio Reporting Summary linked to this article.

## Data availability
All data, protein identifiers (UniProt accession numbers: number E5QVT2, Q1G930, A2RI47, A2RMJ9, P94513, P54595, P0AA93, P0AD14, Q08430, Q2G061, Q9CES5, Q2JYM9, Q488U0, F5RD42, C6WYA8, Q64AC4, E8TNX1, A8TXH3, A0A4R3E8R1, A2SU24) and corresponding protein structures were obtained from online resources publicly available, and described in the main document and the supplementary information. Further details on the analyses carried out in this study are available upon reasonable request to the corresponding author.

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

## Acknowledgements
M.P. thanks Natalia Skinder for the precious assistance with the visuals and constant support, Mark Nijland for insightful discussions, and Prof. Dr. Ciro Leonardo Pierri for valuable remarks on computational protein structure prediction. D.J.S acknowledges funding from the Dutch Research Council: NWO TOP Grant 714.018.003.

## Author contributions
M.P. and D.J.S. conceived the study; M.P. carried out the structural and bioinformatic analyses; M.P. and D.J.S wrote the manuscript.

## Competing interests
The authors declare no competing interests.
