## [Peer Review File · Communications Biology]

Reviewers' comments:

Reviewer #1 (Remarks to the Author):

In this manuscript by Partipilo, M. and Slotboom, DJ. the authors utilize recent developments in AI based protein structure prediction (AlphaFold) and fold recognition (Foldseek) to identify a structural link between the S-component of ECF transporters and the transmembrane region of 5TMR sensor histidine kinases. The manuscript illustrates a beautiful example of how recent developments in AI-based protein structure prediction and bioinformatics resources can provide new and novel insights into common protein structural folds, which could not have been achieved with methods such as protein sequence comparison alone. The finding of the S-component fold being widespread throughout both ECF transporters and histidine kinases in various bacterial phyla highlights the importance of this fold in mediating both signal (nutrient) perception and acquisition.

Overall, this manuscript is very well written and easy to understand. The figures are also very well made and easy to comprehend. I believe that the authors results will be of immense interest to a wide variety of structural biologists, microbiologists, and bioinformaticians. The authors have done an excellent job illustrating the power of AI-based protein structure prediction in facilitating novel biological discoveries with widespread implications across bacterial species.

I have only limited comments for the authors to consider. I do not necessarily think these points must be addressed prior to publication, but simply highlight them for the authors to consider as I believe they may strengthen the already very interesting manuscript...

1. In the manuscript the authors make the claim that their results "manifest the first transceptor case identified in prokaryotes" (line 332-333). I am not sure that this statement is factually accurate, and may largely depend on the authors definition of a "transceptor". For instance...the STAC domain (SLC and TCST-Associated Component) has previously been identified to bridge transmembrane SLC transporter and histidine kinase domains in proteins such as CbrA found in *Pseudomonas* species. In these proteins a single polypeptide contains SLC transporter, STAC, PAS, Dhp, and histidine kinase catalytic domains. Would such a protein not fit within the definition of a "transceptor" identified in prokaryotes? Relevant references below...

Korycinski M, Albrecht R, Ursinus A, Hartmann MD, Coles M, Martin J, Dunin-Horkawicz S, Lupas AN. STAC--A New Domain Associated with Transmembrane Solute Transport and Two-Component Signal Transduction Systems. *J Mol Biol.* 2015 Oct 9;427(20):3327-3339. doi: 10.1016/j.jmb.2015.08.017. Epub 2015 Aug 28. PMID: 26321252.

Wirtz L, Eder M, Schipper K, Rohrer S, Jung H. Transport and kinase activities of CbrA of *Pseudomonas putida* KT2440. *Sci Rep.* 2020 Mar 25;10(1):5400. doi: 10.1038/s41598-020-62337-9. PMID: 32214184

X.X. Zhang, J.C. Gauntlett, D.G. Oldenburg, G.M. Cook, P.B. Rainey. Role of the transporter-like sensor kinase CbrA in histidine uptake and signal transduction *J. Bacteriol.*, 197 (2015), pp. 2867-2878

2. Along similar lines as point 1 above, membrane protein complexes consisting of an ABC transporter and histidine kinase working in tandem to transport molecules and mediate signaling are well documented in bacteria. For instance, the PstSCAB-PhoUR system in *E. coli* is well known to couple phosphate transport and signaling in a single membrane protein complex...

Gardner SG, McCleary WR. Control of the phoBR Regulon in *Escherichia coli*. *EcoSal Plus.* 2019 Sep;8(2). doi: 10.1128/ecosalplus.ESP-0006-2019. PMID: 31520469.

Similarly, the BceAB-RS system of *Bacillus subtilis* consists of an ABC transporter interfacing with a histidine kinase to mediate signaling in response to antimicrobial peptides...

George NL, Orlando BJ. Architecture of a complete Bce-type antimicrobial peptide resistance module. *Nat Commun.* 2023 Jul 1;14(1):3896. doi: 10.1038/s41467-023-39678-w. PMID: 37393310

While such systems are distinct from the S-component containing proteins identified in the current manuscript as they are not confined to a single polypeptide, one could envision them being considered "transceptor complexes". It may be worthwhile to mention proteins like CbrA, and complexes like PstSCAB-PhoUR and BceAB-RS in the discussion to highlight that combined signaling/transport has been documented in bacterial proteins(complexes) other than the S-component proteins identified in the current manuscript.

3. In figure 2A many of the proteins identified to contain an S-component fold do not have annotated function(grey boxes). I am curious, do these proteins without functional annotation tend to have AlphaFold predicted structures that may hint at their function? How many of these non-annotated proteins are also likely to be S-component containing histidine kinases.

4. In the paragraph starting on line 208 the authors describe features of 5TMR SHK helix-0, and suggest that this helix may mediate dimerization. Given the power of AlphaFold-multimer in predicting protein-protein interactions, it seems likely that the authors could rapidly obtain a dimerized model of one of the S-component containing 5TMR-HKs. Such a model may lend credence to their hypothesis that H0 is important in mediating dimerization.

5. Similarly to point 4 above, in the final model proposed in the discussion (line 370) and figure 6 the authors mention how substrate binding to the S-component domain of SHKs may promote their dimerization and initiation of the phosphorelay cascade. Alternatively, one could envision that the SHKs exist predominantly as dimers in the membrane even in the absence of substrate, and substrate binding to the S-component domain would trigger conformational changes that propagate throughout the histidine kinase to initiate signaling. In this reviewer's opinion, the alternate scenario with pre-dimerized SHK may be more consistent with current models of SHK structure/function. However, I acknowledge that more experimental evidence with S-component containing SHKs will be required to delineate between these two hypothetical models.

6. line 119: "the best-scoring matches were homologues S-components"
This wording is a little odd. Maybe "homologues" should be "homologous"?

Reviewer #2 (Remarks to the Author):

The manuscript submitted by Drs Partipilo and Slotboom represent a new bioinformatics analysis of an unique family of microbial transporter proteins, that was attracted attention of many microbiologists, biochemists and structural biologists during the last decade. All ECF transporters contain a substrate specific S-component and many of them share the energy-coupling components termed EcfAA`T, that provide the essential ATP-dependent energy for an unique uptake mechanisms. The tertiary structures of many S-components have been solved during recent years and the authors explored their structural homologs among both existing tertiary structures (from within the PDB Databank) and also from the recently combined new databases containing the AlphaFold model predicted protein structures. Using the bioinformatics tools to search for a structurally similar proteins taht cannot be identified using the protein primary structure similarity search tools (such as BLAST), the authors were able to identify (i) a high structural similarity between various S-components that are specific to different substrates and share only limited protein sequence similarity, and (ii) a significant structural

similarity between S-components and proteins representing families of sensor histidine kinases (their transmembrane domains) and the MreD family of poorly characterized proteins involved in the shaping of bacterial cell. This is a very interesting and important finding that sheds the light into a possible evolutionary origin of these protein families. The authors investigated further the phylogenetic distribution and domain compositions of the studied and identified protein families and provided a few hypotheses on fold specialization of sensor kinases and ECF transporters. The study is well planned, the bioinformatics methods used are solid and the manuscript is well written and I found it easy to follow. Congratulations to the authors for this excellent example of practical use of AlphaFold generated structures for protein evolutionary studies.

Reviewer #3 (Remarks to the Author):

This is a well-written paper that provides support for an interesting hypothesis about evolution, structure, and function of important protein families in bacteria. The strength of this manuscript is in its clarity and interesting results. Weaknesses of this manuscript are the lack of any experimental data. The authors could also more clearly state the level of confidence in their approach. They cite papers related to the strengths/weaknesses of AlphaFold analyses, such as citation #47, but don't elaborate much on why this work is cited.

Fig. 1 - Since the figure is already in color, change EcfA and EcfA' to a different color (shades of yellow, perhaps). I realize that they are quite different, but the cell membrane is also light grey, so it would improve the figure if the EcfA proteins were not grey.

Line 132: awkward sentence structure. Perhaps change to:
most likely because the sole function of MreD is determining cell shape in rod-shaped bacteria

Line 200: The AlphaFold structures of representative proteins with the S-component fold from ECF transporters, SHKs and MreDs were used for structural multi-alignments and build a phylogenetic tree (Figure S1 and Table S5).

Change this to: were used for structural multi-alignments and to build a phylogenetic tree (Figure S1 and Table S5).

Also - I don't think that the trees need to be shown, even as a supplemental figure. As built and shown, they are not very useful.

Fig 3B, 4 and 6 - can this figure be changed to make it simpler for people who can't see red or green? Using brown, purple, and orange shades in place of red and green will make the figure more accessible.

Fig. 3 could be a table - why not just provide the numbers used to make these figures? It is simple for the reader to make a figure if they wish.

Fig 6 is nicely put together and would be useful for teaching and developing further hypotheses about these proteins.

Fig S1. This line in the legend doesn't make sense: The tree scale indicates 31 the number of amino acid substitutions for site.

Do you mean per site?

Reviewers' comments:

Reviewer #1 (Remarks to the Author):

In this manuscript by Partipilo, M. and Slotboom, DJ. the authors utilize recent developments in AI based protein structure prediction (AlphaFold) and fold recognition (Foldseek) to identify a structural link between the S-component of ECF transporters and the transmembrane region of 5TMR sensor histidine kinases. The manuscript illustrates a beautiful example of how recent developments in AI-based protein structure prediction and bioinformatics resources can provide new and novel insights into common protein structural folds, which could not have been achieved with methods such as protein sequence comparison alone. The finding of the S-component fold being widespread throughout both ECF transporters and histidine kinases in various bacterial phyla highlights the importance of this fold in mediating both signal (nutrient) perception and acquisition.

Overall, this manuscript is very well written and easy to understand. The figures are also very well made and easy to comprehend. I believe that the authors results will be of immense interest to a wide variety of structural biologists, microbiologists, and bioinformaticians. The authors have done an excellent job illustrating the power of AI-based protein structure prediction in facilitating novel biological discoveries with widespread implications across bacterial species.

We thank the reviewer for the kind words on our work.

I have only limited comments for the authors to consider. I do not necessarily think these points must be addressed prior to publication, but simply highlight them for the authors to consider as I believe they may strengthen the already very interesting manuscript...

1. In the manuscript the authors make the claim that their results "manifest the first transceptor case identified in prokaryotes" (line 332-333). I am not sure that this statement is factually accurate, and may largely depend on the authors definition of a "transceptor". For instance...the STAC domain (SLC and TCST-Associated Component) has previously been identified to bridge transmembrane SLC transporter and histidine kinase domains in proteins such as CbrA found in *Pseudomonas* species. In these proteins a single polypeptide contains SLC transporter, STAC, PAS, DHp, and histidine kinase catalytic domains. Would such a protein not fit within the definition of a "transceptor" identified in prokaryotes? Relevant references below...

Korycinski M, Albrecht R, Ursinus A, Hartmann MD, Coles M, Martin J, Dunin-Horkawicz S, Lupas AN. STAC--A New Domain Associated with Transmembrane Solute Transport and Two-Component Signal Transduction Systems. *J Mol Biol.* 2015 Oct 9;427(20):3327-3339. doi: 10.1016/j.jmb.2015.08.017. Epub 2015 Aug 28. PMID: 26321252.

Wirtz L, Eder M, Schipper K, Rohrer S, Jung H. Transport and kinase activities of CbrA of *Pseudomonas putida* KT2440. *Sci Rep.* 2020 Mar 25;10(1):5400. doi:

10.1038/s41598-020-62337-9. PMID: 32214184

X.X. Zhang, J.C. Gauntlett, D.G. Oldenburg, G.M. Cook, P.B. Rainey. Role of the transporter-like sensor kinase CbrA in histidine uptake and signal transduction J. Bacteriol., 197 (2015), pp. 2867-2878

We agree with the reviewer that the definition of transceptor used in this manuscript requires clarification, as several cases of small domains or full-length proteins bridging signal sensing and membrane transport are already known in prokaryotes. We now have clarified in our Introduction (lines 48-52):

'Here, we use a narrow definition of transceptors as integral membrane proteins that structurally resemble transporters, yet function as receptors. This definition excludes cases in which a dedicated soluble domain acts as a bridge between transport proteins and signal transduction systems (e.g. STAC domain alone or incorporated into CbrA from *Pseudomonas putida*).^{10,11}

2. Along similar lines as point 1 above, membrane protein complexes consisting of an ABC transporter and histidine kinase working in tandem to transport molecules and mediate signaling are well documented in bacteria. For instance, the PstSCAB-PhoUR system in *E. coli* is well known to couple phosphate transport and signaling in a single membrane protein complex...

Gardner SG, McCleary WR. Control of the phoBR Regulon in *Escherichia coli*. EcoSal Plus. 2019 Sep;8(2). doi: 10.1128/ecosalplus.ESP-0006-2019. PMID: 31520469.

Similarly, the BceAB-RS system of *Bacillus subtilis* consists of an ABC transporter interfacing with a histidine kinase to mediate signaling in response to antimicrobial peptides...

George NL, Orlando BJ. Architecture of a complete Bce-type antimicrobial peptide resistance module. Nat Commun. 2023 Jul 1;14(1):3896. doi: 10.1038/s41467-023-39678-w. PMID: 37393310

While such systems are distinct from the S-component containing proteins identified in the current manuscript as they are not confined to a single polypeptide, one could envision them being considered "transceptor complexes". It may be worthwhile to mention proteins like CbrA, and complexes like PstSCAB-PhoUR and BceAB-RS in the discussion to highlight that combined signaling/transport has been documented in bacterial proteins(complexes) other than the S-component proteins identified in the current manuscript.

We thank the reviewer for these remarks. In order not to lose the focus on AI-predictions and S-components in our Discussion section, we decided to mention well-established examples of interactions between transporters and receptors in our Introduction instead.

BceAB-S is used to exemplify the formation of complexes occurring between transporters and receptors, together with DctA-DcuS from *Escherichia coli* (10.1111/j.1365-2958.2012.08143.x). This is stated as follows (lines 41-46):

Although mediated by distinct protein components, transport and signal sensing can synergistically take place within larger complexes, as demonstrated by the C4-dicarboxylate transporter DctA that forms a complex with the fumarate sensor DcuS in *Escherichia coli*,⁵ and the structurally characterized BceAB-S module from *Bacillus subtilis*,⁶ where an ABC transporter (BceAB) interfaces an histidine kinase (BceS) to respond to antimicrobial peptides.'

3. In figure 2A many of the proteins identified to contain an S-component fold do not have annotated function(grey boxes). I am curious, do these proteins without functional annotation tend to have AlphaFold predicted structures that may hint at their function? How many of these non-annotated proteins are also likely to be S-component containing histidine kinases.

Most of the proteins resulting from the Foldseek search are without functional annotation in the related UniProt entry. They are likely S-components without functional annotation. This is now explicitly stated in the Results section (lines 144-146):

Finally, entries not functionally annotated represented 20 and 60% of the hits, depending on the specific search, which in most cases are likely to be S-components (although not annotated as such in protein databases).'

Only a marginal number of the none-annotated protein hits (mostly close homologues to each other) are 5TMR-SHKs. Below we attach a table of representative results from the SWISSPROT Foldseek search (with a larger number of hits) supporting this conclusion, in which repetitions among different search-results are omitted.

AlphaFold structure	Amino acids	UniProt annotation	Organism	Potential function	Query Structure
AF-P75251-F1-model_v4	225	Uncharacterized protein MG350.1 homolog	Mycoplasmoides pneumoniae M129	Unknown	3P5N
AF-C4L1J3-F1-model_v4	183	UPF0397 protein EAT1b_2102	Exiguobacterium sp. AT1b	S-component	3P5N
AF-P47393-F1-model_v4	375	Uncharacterized protein MG147	Mycoplasmoides genitalium G37	Unknown	3P5N
AF-Q49VU5-F1-model_v4	356	Uncharacterized membrane protein SSP1970	Staphylococcus saprophyticus subsp. saprophyticus ATCC 15305 = NCTC 7292	5TMR-SHK	3P5N
AF-Q87G36-F1-model_v4	182	UPF0397 protein VPA1481	Vibrio parahaemolyticus RIMD 2210633	S-component	3P5N
AF-Q8CTF5-F1-model_v4	356	Uncharacterized membrane protein SE_0528	Staphylococcus epidermidis ATCC 12228	5TMR-SHK	3P5N
AF-Q58333-F1-model_v4	297	Uncharacterized protein MJ0923	Methanocaldococcus jannaschii DSM 2661	Unknown	3P5N
AF-Q9CIN2-F1-model_v4	182	UPF0397 protein YdcD	Lactococcus lactis subsp. lactis II1403	S-component	AF-E5QVT2-F1
AF-Q5M5Y6-F1-model_v4	183	UPF0397 protein stu0306/stu0307	Streptococcus thermophilus LMG 18311	S-component	AF-E5QVT2-F1
AF-B1V944-F1-model_v4	193	UPF0397 protein PA0141	Candidatus Phytoplasma australiense	S-component	5D0Y
AF-A8FJA6-F1-model_v4	185	UPF0397 protein BPUM_3679	Bacillus pumilus SAFR-032	S-component	5D0Y
AF-Q03US7-F1-model_v4	187	UPF0397 protein LEUM_1974	Leuconostoc mesenteroides subsp. mesenteroides ATCC 8293	S-component	5D0Y
AF-Q88Z21-F1-model_v4	186	UPF0397 protein lp_0150	Lactiplantibacillus plantarum WCFS1	S-component	5D0Y
AF-Q7A341-F1-model_v4	184	UPF0397 protein SA2477	Staphylococcus aureus subsp. aureus N315	S-component	AF-Q1G930-F1
AF-B2IM88-F1-model_v4	182	UPF0397 protein SPCG_0463	Streptococcus pneumoniae CGSP14	S-component	AF-Q1G930-F1

AF-P22821-F1-model_v4	166	Protein BioX	Lysinibacillus sphaericus	S-component/Unknown	AF-Q1G930-F1
AF-Q832R4-F1-model_v4	183	UPF0397 protein EF_2154	Enterococcus faecalis V583	S-component	4POP
AF-Q5HHS4-F1-model_v4	356	Uncharacterized membrane protein SACOL0809	Staphylococcus aureus subsp. aureus COL	5TMR-SHK	4POP
AF-A4VTD2-F1-model_v4	181	UPF0397 protein SSU05_0404	Streptococcus suis 05ZYH33	S-component	AF-A2RI47-F1
AF-C3LH78-F1-model_v4	182	UPF0397 protein BAMEG_1951	Bacillus anthracis str. CDC 684	S-component	AF-A2RI47-F1
AF-Q5E4I5-F1-model_v4	183	UPF0397 protein VF_1566	Aliivibrio fischeri ES114	S-component	AF-A2RI47-F1
AF-Q2FIP5-F1-model_v4	356	Uncharacterized membrane protein SAUSA300_0730	Staphylococcus aureus subsp. aureus USA300	5TMR-SHK	AF-A2RI47-F1
AF-P48331-F1-model_v4	188	Uncharacterized 21.2 kDa protein in psbX-ycf33 intergenic region	Cyanophora paradoxa	S-component	4DVE
AF-B3W705-F1-model_v4	186	UPF0397 protein LCABL_04350	Lacticaseibacillus casei BL23	S-component	4DVE
AF-B9DTI6-F1-model_v4	181	UPF0397 protein SUB0313	Streptococcus uberis O140J	S-component	4DVE
AF-Q6GDB9-F1-model_v4	184	UPF0397 protein SAR2767	Staphylococcus aureus subsp. aureus MRSA252	S-component	4DVE
AF-P20298-F1-model_v4	144	Uncharacterized protein in gap 3'region	Pyrococcus woesei	S-component/Unknown	4DVE
AF-Q7MFH2-F1-model_v4	182	UPF0397 protein VVA0348	Vibrio vulnificus YJ016	S-component	AF-A2RMJ9-F1
AF-A9VHJ1-F1-model_v4	182	UPF0397 protein BcerKBAB4_2500	Bacillus mycoides KBAB4	S-component	AF-A2RMJ9-F1
AF-Q2YSF9-F1-model_v4	356	Uncharacterized membrane protein SAB0698c	Staphylococcus aureus RF122	5TMR-SHK	AF-A2RMJ9-F1
AF-Q49414-F1-model_v4	287	Uncharacterized protein MG313	Mycoplasmoides genitalium G37	Unknown	AF-A2RMJ9-F1

4. In the paragraph starting on line 208 the authors describe features of 5TMR SHK helix-0, and suggest that this helix may mediate dimerization. Given the power of AlphaFold-multimer in predicting protein-protein interactions, it seems likely that the authors could rapidly obtain a dimerized model of one of the S-component containing 5TMR-HKs. Such a model may lend credence to their hypothesis that H0 is important in mediating dimerization.

We thank the reviewer for the suggestion. We now modified Figure 3A, including an AlphaFold multimeric prediction of YpdA highlighting the AI-predicted interactions at the level of helix0. We modified the text accordingly (lines 232-235):

AlphaFold predictions for the homodimeric conformation of 5TMR-SHKs highlight the close proximity – and probable interaction - of the conserved polar side chains (around 4 Å), corroborating the hypothesis of the formation of an intramembranous salt bridge between two protomers (Figure 4A).

A dedicated paragraph describing the procedure behind the AlphaFold-prediction generated via ColabFold is available in the Material and Methods section (lines 476-481).

5. Similarly to point 4 above, in the final model proposed in the discussion (line 370) and figure 6 the authors mention how substrate binding to the S-component domain of SHKs may promote their dimerization and initiation of the phosphorelay cascade. Alternatively, one could envision that the SHKs exist predominantly as dimers in the membrane even in the absence of substrate, and substrate binding to the S-component domain would trigger conformational changes that propagate throughout the histidine kinase to initiate signaling. In this reviewer's opinion, the alternate

scenario with pre-dimerized SHK may be more consistent with current models of SHK structure/function. However, I acknowledge that more experimental evidence with S-component containing SHKs will be required to delineate between these two hypothetical models.

We thank the reviewer for raising the point about the preexisting dimeric state, regardless of substrate binding, and have now modified the text (lines 246-251 and 384-398):

While in ECF transporters, the binding of substrate leads to association with the ECF module, we speculate that in SHKs, conformational changes upon substrate binding may either cause dimerization of monomeric receptors, or a specific reorganization of pre-existing interactions in the dimeric receptor complex, which subsequently leads to the transmission of the signal from the membrane to the soluble domains.

and

Rather than toppling and associating with the ECF module, potential conformational changes at the membrane domain interface induced by the substrate recognition may culminate into an orchestrated series of intramolecular rearrangements of the cytosolic domains, and eventually enable the downstream phosphorelay cascade.

Thus, the cartoon in figure 6 has been modified accordingly.

6. line 119: "the best-scoring matches were homologues S-components"
This wording is a little odd. Maybe "homologues" should be "homologous"?

We thank the reviewer for noticing the odd wording and changed 'homologues S-components' into 'homologous S-components'.

Reviewer #2 (Remarks to the Author):

The manuscript submitted by Drs Partipilo and Slotboom represent a new bioinformatics analysis of an unique family of microbial transporter proteins, that was attracted attention of many microbiologists, biochemists and structural biologists during the last decade. All ECF transporters contain a substrate specific S-component and many of them share the energy-coupling components termed EcfAA`T, that provide the essential ATP-dependent energy for an unique uptake mechanisms. The tertiary structures of many S-components have been solved during recent years and the authors explored their structural homologs among both existing tertiary structures (from within the PDB Databank) and also from the recently combined new databases containing the AlphaFold model predicted protein structures. Using the bioinformatics tools to search for a structurally similar proteins that cannot be identified using the protein primary structure similarity search tools (such as BLAST), the authors were able to identify (i) a high structural similarity between various S-components that are specific to different substrates and share only limited protein sequence similarity, and (ii) a significant structural similarity between S-components and proteins representing families of sensor histidine

kinases (their transmembrane domains) and the MreD family of poorly characterized proteins involved in the shaping of bacterial cell. This is a very interesting and important finding that sheds the light into a possible evolutionary origin of these protein families. The authors investigated further the phylogenetic distribution and domain compositions of the studied and identified protein families and provided a few hypotheses on fold specialization of sensor kinases and ECF transporters. The study is well planned, the bioinformatics methods used are solid and the manuscript is well written and I found it easy to follow. Congratulations to the authors for this excellent example of practical use of AlphaFold generated structures for protein evolutionary studies.

We thank the reviewer for the flattering words about our overall work, in terms of the used methods, clarity in writing, up to the potential scientific output.

Reviewer #3 (Remarks to the Author):

This is a well-written paper that provides support for an interesting hypothesis about evolution, structure, and function of important protein families in bacteria. The strength of this manuscript is in its clarity and interesting results. Weaknesses of this manuscript are the lack of any experimental data. The authors could also more clearly state the level of confidence in their approach. They cite papers related to the strengths/weaknesses of AlphaFold analyses, such as citation #47, but don't elaborate much on why this work is cited.

We appreciate the reviewer's description of the strengths to our manuscript and agree with the comments on the lack of experimental data. We now explicitly state that experimental work is required in future (lines 355-356):

"While it is important that the predicted structural relatedness between 5TMR-SHKs and S-components will be tested experimentally in future work⁵³,..."

Nonetheless, the main aim of our work was providing a new structural AI-driven approach for the development of biological hypotheses, that would otherwise be impossible using the conventional sequence similarity-based tools. This work alone required a full manuscript, and we feel that our analyzes have served this purpose, with the hope of inspiring scientists from diverse backgrounds to look for hitherto unknown connections in other protein families, and subsequently test them experimentally.

In the absence of experimental evidence, we are not currently able to absolutely prove or disprove the hypotheses, and we strongly rely on AlphaFold predictions that have limitations described in detail in reference 47 (now ref. 53). We now provide a clearer explanation of these limitations, while stating more in detail the level of confidence in our approach in the Discussion section (lines 356-366):

several indicators show that the confidence of the prediction is high: i) with both AI-predicted and experimentally determined structures of S-components used as queries structural homology between S-components and 5TMR-SHKs was found (Fig. 2A), ii) the confidence (TM) scores obtained for the hits were high, in almost all cases above 0.5 (Table S2), iii) the 5TMR-SHK hits were interspersed in the ranking among hits of validated S-components, iv) fold recognition from the sequence of the membrane domain of the 5TMR-SHKs always found crystallographic structures of S-components (Tables S3-S4), v) well conserved amino acids in the pocket for substrate recognition in 5TMR-SHKs (Fig. 4B) correspond to amino acids involved in substrate binding in S-components with different specificities (Fig S4).

Fig. 1 - Since the figure is already in color, change EcfA and EcfA' to a different color

(shades of yellow, perhaps). I realize that they are quite different, but the cell membrane is also light grey, so it would improve the figure if the EcfA proteins were not grey.

We understand the concern of the reviewer about too many elements in grey, and thus decided to change the color of the membrane cartoon to avoid confusion. Consistently, we did the same with figure 6.

Line 132: awkward sentence structure. Perhaps change to:
most likely because the sole function of MreD is determining cell shape in rod-shaped bacteria

To improve the sentence structure, we rephrased it into 'because the sole function of MreD is tightly coupled to determining cell architecture in rod-shaped bacteria', ensuring to avoid bold statements on the precise function of MreD in the complex process of shape determination of rod bacteria.

Line 200: The AlphaFold structures of representative proteins with the S-component fold from ECF transporters, SHKs and MreDs were used for structural multi-alignments and build a phylogenetic tree (Figure S1 and Table S5).
Change this to: were used for structural multi-alignments and to build a phylogenetic tree (Figure S1 and Table S5).

We rephrased line 200 as suggested.

Also - I don't think that the trees need to be shown, even as a supplemental figure. As built and shown, they are not very useful.

The reviewer's point here is understandable, since we (authors) carefully considered before submitting the article whether it would be appropriate to show these results in the manuscript. Our aim with Figure S1 was to highlight the limitations of a structural alignment built on entries that are extremely different in sequence and function, regardless of the predictive or experimental nature of the protein structures. We believe it is very important to understand whether this shared fold between transport proteins, receptors and scaffold proteins (proposed function of MreD) had a common origin, or is the result of independent evolutionary events in the fold landscape of membrane proteins. While at present we have not been able to answer this intriguing question, even including distinct analyzes with only the membrane domain or full-length 5TMR-SHKs, we hope that our efforts represented in Figure S1 can be the starting point for further research. We prefer to keep the figure, but leave it to the editor to decide.

Fig 3B, 4 and 6 - can this figure be changed to make it simpler for people who can't

see red or green? Using brown, purple, and orange shades in place of red and green will make the figure more accessible.

We agree, and to make our figures more accessible, we modified figures 4 and 6

Fig. 3 could be a table - why not just provide the numbers used to make these figures? It is simple for the reader to make a figure if they wish.

We believe that this is a matter of taste. We prefer figures over table. Furthermore, the numbers are continuously being updated with new InterPro releases (on average there are four new releases every year), and would soon become partially inaccurate, while the pie charts and bar diagrams will by-and-large will remain unchanged. The bar graph in Figure 3A and the pie graph in Figure 3B make readers immediately aware of the relationships between S-components, 5TMR-receptors and MreD proteins in terms of taxonomic distribution, regardless of what the absolute numbers are. For this reason, we believe the choice of visual representation of the data is more suitable to the test of time and to the constant update of database annotations, without compromising the take-home message we want to convey to the readership.

For information we added the tables here:

	Bacterial species (%)	Archaeal species (%)	Not assigned species (%)
S-components	98.4	1.5	0.1
5TMR-SHKs	97.6	2.2	0.2
MreDs	99.9	0.0	0.1

Table 1 The occurrence among different species of the proteins with the shared fold. Data was obtained from the respective PFAM entries available in the 'Taxonomy' section of the InterPro database <https://www.ebi.ac.uk/interpro/entry/pfam/> (PF12822 for S-components, PF04093 for MreD proteins and PF07694 for 5TMR-containing SHKs).

	Bacillota	Actinomy cetota	Mycoplas matota	Pseudom onatota	Thermoto gota	Spirochae tota	Thermode sulfobact eriotota	Others	Total species
S- compo nents	4249	1276	133	457	55	53	20	366	6609
5TMR- SHKs	2088	11	31	2358	33	40	148	432	5141
MreDs	4212	1098	5	6981	0	239	96	1588	14219

Table 2 The distribution of S-components, SHKs and MreDs among bacterial phyla. The analysis on the number of species refers to the release InterPro 96.0.

Fig 6 is nicely put together and would be useful for teaching and developing further hypotheses about these proteins.

We thank the reviewer for the kind words..

Fig S1. This line in the legend doesn't make sense: The tree scale indicates 31 the number of amino acid substitutions for site.
Do you mean per site?

We indeed meant substitution per site, and modified it accordingly.

REVIEWERS' COMMENTS:

Reviewer #4 (Remarks to the Author):

The response seems convincing and the proposed changes, in my opinion, suitably address R3's concerns. Of course, the authors have not added experimental work but I agree with them that this seems to go beyond the goal of this work, which was to demonstrate the use of AI based prediction for identifying conserved folds in distantly related proteins.